# The Competitive Mating of Irradiated Brown Marmorated Stink Bugs, *Halyomorpha halys*, for the Sterile Insect Technique

**DOI:** 10.3390/insects10110411

**Published:** 2019-11-16

**Authors:** David Maxwell Suckling, Massimo Cristofaro, Gerardo Roselli, Mary Claire Levy, Alessia Cemmi, Valerio Mazzoni, Lloyd Damien Stringer, Valeria Zeni, Claudio Ioriatti, Gianfranco Anfora

**Affiliations:** 1The New Zealand Institute for Plant and Food Research Ltd, PB 4704, Christchurch 8140, New Zealand; Lloyd.Stringer@plantandfood.co.nz; 2School of Biological Sciences, University of Auckland, Auckland 1072, New Zealand; 3Technology Transfer Center, Fondazione Edmund Mach, I-38010 San Michele all’Adige, Italy; gerardoroselli@hotmail.it (G.R.); claudio.ioriatti@fmach.it (C.I.); 4Italian National Agency for New Technologies, Energy and Sustainable Economic Development (ENEA), 00123 Rome, Italy; m.cristofaro55@gmail.com (M.C.); alessia.cemmi@enea.it (A.C.); 5Biotechnology and Biological Control Agency (BBCA onlus), 00123 Rome, Italy; clairelevy@xtra.co.nz; 6Center of Agriculture, Food and Environment (C3A), University of Trento, I-38010 San Michele all’Adige, Italy; gianfranco.anfora@fmach.it; 7Kallisto, Christchurch 8081, New Zealand; 8Research and Innovation Center, Fondazione Edmund Mach, I-38010 San Michele all’Adige, Italy; valerio.mazzoni@fmach.it (V.M.); valeriazeni93@gmail.com (V.Z.)

**Keywords:** irradiation, stink bug, sterile insect technique, suppression, sterility, *Halyomorpha halys*, SIT, wild harvest

## Abstract

The sterility of eggs and nymphs from gamma-irradiated male *Halyomorpha halys* was investigated to determine the potential for the sterile insect technique (SIT). Males irradiated at 0, 16, 24 and 32 Gy were placed with untreated virgin females, and egg sterility was determined, showing 54.3% at 16 Gy. The percentage of sterility from irradiation was 26 percent lower than previous results from the USA and the variance was very high. Competitive overflooding ratio trials between irradiated virgin males and fertile virgin males at a 5:1 ratio resulted in the expected egg sterility, indicating competitive performance by irradiated males. By July and August, older, irradiated overwintered males were significantly less competitive than similar, non-irradiated males. There is a need to revisit the irradiation delivery method to achieve proper precision around the paternal dose required for an expected >80% egg sterility and subsequent ~99% endpoint sterility estimated at adult emergence in the F_1_ phase. These results suggest that the mating competitiveness and competency of males after irradiation at 16 Gy is not limiting to the sterile insect technique for suppression. A wild harvest of overwintering males using the aggregation pheromone, followed by irradiation and male release, might replace rearing. Mass-collected, sterilized bugs could be transported from an area of high *H. halys* density and shipped for release to enable suppression or eradication elsewhere. This concept is under development but further work is needed now to understand the difference in results between the US and Italian irradiators and increase the reliability of dosimetry.

## 1. Introduction

*Halyomorpha halys* Stål (Hemiptera: Pentatomidae), the brown marmorated stink bug (BMSB), is a polyphagous multivoltine pest from eastern Asia [1]. The invaded areas include Chile, the USA and Europe [2,3] where many economically-important crops are affected [4]. Additional problems are caused by this pest during overwintering diapause, since it aggregates in man-made structures, making it a pervasive urban nuisance [4,5]. *Halyomorpha halys* has been targeted by a range of insecticides [6], but more benign and targeted methods for control would be desirable [7,8,9,10]. Surveillance trapping may be possible using an aggregation pheromone [11,12], but suppression with insecticides is difficult and eradication has not been attempted so far.

The sterile insect technique (SIT), one of various strategies to achieve control of insect pests, is capable of achieving eradication, if certain conditions are met [13,14]. The SIT is species-specific and has no off-target effects, but the irradiation biology of Hemiptera has been poorly studied so far and there have been no field applications of SIT. Studies have mainly focused on the determination of effective sterilizing doses [8,15,16], and determining the most radio-resistant life stage [17,18]. Stringer et al. [19] reported several effects of irradiation on spermatogenesis in *Nezara viridula* L., including the formation of beta chromosomes, which have some potential for transgenerational mortality, contributing to SIT. The consequences of increasing the frequency of beta chromosomes could be deleterious for up to three generations, according to work on milkweed bug (*Oncopeltus*), [20], but the contribution of this effect on suppression is unclear. While there is a positive relationship between mating frequency and fecundity [21], as long as the overflooding ratio is suitably maintained, the success of sterile release should not be adversely affected by multiple mating [21,22].

The competitive fitness of sterile and wild insects is critical for success, and often short-term mating experiments can be used to determine whether sterile insects participate proportionally [23,24,25]. Multiple mating and longevity of weeks by BMSB meant that we needed another approach to assess fitness, so we compared the level of egg sterility observed from cages, with an over-flooding ratio.

A recent investigation [26] into the effect of radiation on male *H. halys* in the context of the sterile insect technique reported a dose of 16 Gray (Gy) to adult males to confer 80% sterility on eggs resulting from mating with virgin, non-irradiated females. The subsequent mortality of developing nymphs and F_1_ adults accumulated close to 99% mortality by the point of F_2_ egg hatch. No lasting effect from irradiation was observed in the subsequent generation. It appears that only conventional SIT is possible in this species; i.e., sterility is not inherited to a sufficient extent by the subsequent generation. A current study in Italy used another irradiator and strain of *H. halys* at a single dose for comparison of the egg sterility as an endpoint bioassay, based on Welsh et al. from the USA [26]. Ultimately, we sought to determine the competitive ability of irradiated adult male *H. halys*, in the context of evaluating the potential application of the SIT for use in an eradication or suppression program.

## 2. Materials and Methods

Rearing. Cloth cages (30 × 30 × 30 cm, Bugdorm, Taiwan) were set up at Fondazione Edmund Mach, San Michele, with paper towels on the floor and paper suspended from the ceiling, which successfully attracted oviposition for the assessment of hatch rates. The hatch rate of eggs was determined using a light microscope throughout. Insects were field-collected as overwintered adults and next generation nymphs in June and July 2018 in Friuli Venezia Giulia, Italy (4559′57.5″ N, 13°0′ 38.6″ E) and reared on fresh fruits and vegetables (beans, tomatoes, apples and kiwifruit). Nymphs were isolated at the fifth instar in separate cages to ensure the availability of virgin adults. It was assumed that field-collected adults were likely to have previously mated after coming out from winter diapause. We tested the effect of SIT in a scenario whereby it was likely that unmated individuals would be collected from the field in autumn for the overwintering, chilling, sorting and irradiation of males, and release up to several months later.

Irradiation. Mature, 1–2 week-old, virgin male adults were separated and irradiated in 9 cm diameter cardboard tubes with 16, 24 and 32 Gy at the ^60^Co gamma facility at ENEA (Italian National Agency for New Technologies, Energy and Sustainable Economic Development), Rome [27]. The dose rate was 175.03 Gy/h (2.92 Gy/min) and the dose absorbed was 16.07 Gy, with a theoretical ± 3.5% of difference in terms of irradiation dose for the bugs that were at the closest and/or farthest positions on the inside walls. The dose of 16 Gy was reported to result in 80% sterility at F_1_ egg hatch and 98.5% sterility at F_2_ egg hatch [26]. Consequently, two experiments using 16 Gy were conducted in 2018 that were designed to permit competition between treated and untreated males, based on expectations from the US dosimetry. While eggs from untreated females placed with irradiated males were expected to be the particular value of 80% sterile at hatch [26], after an apparent difference from this expected value in 2018, accompanied by an unexpected scatter in results, the dosimetry presented in Experiment 1 was conducted in 2019.

### 2.1. Experiment 1. Male Dose Response

This experiment was conducted in 2019 in order to explain unexpected results from a 2018 study in Italy. Our Italian results did not align with dose-sterility responses reported b7 a previous collaboration in the USA [26] and a wide scatter was observed in egg sterility in 2018 (labelled below as Experiment 3). The egg sterility assessment was repeated at three doses of irradiation, reported earlier [26]. Ten replicate 1 L plastic cages of two virgin females and four virgin males were established with males irradiated at 0, 16, 24 and 32 Gy at ENEA, Rome in July 2019 [27]. Insects were held in cardboard tubes with 9 cm diameters for irradiation, and dosimetry was supplied by ENEA at a reported specification of ±3.5% per dose. Insects were transported back to San Michele and cages established for egg collection twice weekly. Eggs were allowed to hatch, and the percentage of hatching was recorded. Control mortality was hypothesized to be low and sterility was to increase with dose with a wide scatter (based on scatter observed in 2018).

### 2.2. Experiment 2. Overflooding

The following treatments (irradiated as above) were established in July 2018 using (a) five-to-one overflooding ratios, with five virgin males irradiated at 16 Gy to one virgin, non-irradiated male, placed with five (new season) virgin females; (b) five virgin non-irradiated males with five virgin females as an untreated control; (c) five-to-one ratios with five overwintered irradiated males to one overwintered non-irradiated male with five virgin females; (d) five overwintered, non-irradiated males with five virgin females. Three replicates of each treatment were prepared on 12 July 2018 and adult mortality and egg hatch was determined for each treatment until 2 September 2018. Control mortality was hypothesized to be low and egg sterility to directly reflect the competition of 5:1 sterile to untreated males. This experiment overlooked the need to check the assumed dosimetry and variance in sterility, which was done subsequently.

### 2.3. Experiment 3. Overflooding and Competition

The following treatments (irradiated as above) were established in July 2018 with (a) five-to-one overflooding ratios, with fifteen virgin (new season) irradiated males to three virgin non-irradiated males, and fifteen virgin females; (b) fifteen virgin irradiated males and fifteen virgin females; (c) fifteen virgin non-irradiated males and fifteen virgin females as a control. Three replicates of each treatment were prepared on 24 July 2018. Adult mortality and egg hatch were determined for each treatment, daily, until 2 September 2018. Control mortality was hypothesized to be low and egg sterility to directly reflect the competition of 5:1 sterile to untreated males. Multiple mating was expected to play a role such that there would be the same average value but possibly an increased scatter. This experiment overlooked the need to check the assumed dosimetry and variance in sterility, which was done subsequently.

### 2.4. Statistical Methods

Trends in egg batch sterility were examined using a chi-squared likelihood ratio test for goodness of fit of the observed and expected values in Experiments 2 and 3 [28]. Expected values in Experiments 2 and 3 were calculated from the dosimetry at 16 Gy in Experiment 1 (54% hatch) and the proportion of treated and untreated males in the total pool (1 wild expected to have 100% hatch in progeny and 5 irradiated males expected to have 54% egg fertility in progeny, 5/6 × 54% + 1/6 × 100%, or 62% fertility expected). The hatch data were subjected to an Abbott’s correction to account for the role of control hatch rate on results [29]. Linear regression was used to investigate changes in the sterility of egg batches being laid over time (Experiment 3).

## 3. Results

### 3.1. Experiment 1

Eggs of *Halyomorpha halys* were obtained from crosses between non-irradiated males and virgin females, and crosses between virgin females and males irradiated at 16 Gy (Figure 1).

The spread of levels of sterility per egg batch was unexpectedly broad (at 16 Gy, 7–96% egg hatch), and much lower average sterility than expected [26] was recorded subsequent to 16, 24 and 32 Gy dosages (Figure 2). However, despite the 30% deviation of the mean value, this 2019 data set aligns with the 2018 experiments from the same Italian irradiator, with a 54% mean egg hatch at 16 Gy (Table 1); this scatter explains the scatter in the results in Experiments 2 and 3, which also showed a wide range of values (Figure 3). The greater scatter adds only variance to the range of values and does not affect the trend, which is the key component for judging fitness in a competitive mating scenario.

### 3.2. Experiment 2. Overflooding with Older and Virgin Males

A comparison of egg sterility with overflooding of fertile males by irradiated males showed that egg sterility, as indicated by no hatching (Figure 1), was significantly higher in virgin females crossed with irradiated males, compared with the virgin females mated crossed with untreated control males (Table 2). The effect of control mortality was factored into the expected values [29]. Results trended the same way for new season virgins and for overwintered, mated males but the competitiveness of the latter was apparently more affected by irradiation. Mortality of virgin and older adults in culture was similar, about one per two days for both sexes, but was generally higher than controls (see [25]). In the case of competing irradiated virgin males, the expected and observed egg hatch rates were lower than the expected value based on 54% fertility at 16 Gy and 5:1 overflooding (49 versus 62%). A 13% lower egg hatch than expected could indicate greater sterility from this experiment than expected from the ratio of 5:1 gamma-treated males, but that is of a similar size to the control mortality here, so it could probably be best explained as in the range expected. For the considerably older overwintered males, there appeared to be a greater effect of irradiation on male mating success, with less competitive mating from older irradiated males (expected compared with more sterility than expected in virgin males). There was less mortality in the hatching of the controls of the older males (6% versus 15% in virgins).

### 3.3. Experiment 3. Overflooding and Egg Sterility over Time

Adults and eggs were monitored daily across the three treatments to determine trends, and the observed hatch pooled from the treatment with 5:1 irradiated males was not significantly different from the expected value of 62% (Table 3). The mortality of eggs (Figure 3) from the control group was generally low and in the same range as in Experiment 2 (2019). The irradiated male × virgin female crosses (39.7% hatch, Table 3) produced less hatching eggs than expected (54%, from Table 1, *p* < 0.001).

## 4. Discussion

New proposed targets of SIT emerge slowly, against a background where the SIT has been well developed and deployed against only a small range of pest species of Diptera and Lepidoptera [23]. Despite radiation biology being known regarding a large range of species in these orders, the results have not been translated into many field programs [30,31,32]. The best-known examples of large programs, such as new world screw-worm *Cochliomyia hominivorax* and Mediterranean fruit fly *Ceratitis capitata,* have been supplemented by smaller programs on tsetse flies, mosquitoes and fruit fly species, including the Queensland fruit fly *Bactrocera tryoni* [33], the oriental fruit fly *B. dorsalis* [34] and the melon fly *B. cucurbitae* [35]. Programs supported by large scale factory production of pink bollworm *Pectinophora gossypiella* and codling moth *Cydia pomonella* in the USA and Canada have likewise been supplemented by a small group of other programs against moth pests, including for use in eradication [32]. While a range of reasons can be enumerated that explain the slow development of SIT, in general there is the need for a basic economic case for the construction of a factory, a high threshold of investment to reach. However, boutique SIT should not be overlooked, especially for eradication, when there are low-density field populations of the pest [36]. Newly proposed targets for SIT in novel orders, such as the brown marmorated stink bug from the order Hemiptera, require facing and overcoming the same steep hurdles of supply and logistics if SIT is to succeed, but there may be other solutions than those used in conventional SIT programs.

Most eggs exhibited either no development or hatched successfully, although a small proportion of developed embryos died before eclosion. Control mortality was very low initially, which enabled a more accurate assessment of the effects of irradiation. Overwintered males were aging at the time of Experiment 2 (July), so apparently suffered greater effects of irradiation, affecting their competitiveness compared with virgin males from the culture. Because the egg hatch from females mated to irradiated fathers showed lower hatch rates than the untreated control, it is possible to see that irradiated males were competitive with the untreated males because the incidence of sterile eggs was higher than expected. Importantly, lower hatch rates than expected, based on research presented by Welsh et al. [26], were observed with similarly-treated virgin male parents.

Fortunately, the exact values do not affect the interpretation of these results on mating competition, which appears promising from all the <54% hatch values in the 5:1 treatment. While it would have been desirable to have much less or no scatter, this was not possible to know in advance in 2018. For the future, there is a requirement to know the full endpoint sterility and its variance at whatever dose is chosen that is likely to avoid contributing to a field population, and this should be confirmed for any insects prior to release. For now, results pivot around 54%, ensuring that the competition experiment has maximum resolution on either side of this value, which is suitable here because there is the maximum range for the control (expected to be low) and there is the expectation of near 54% values for the SIT treatment (~62%).

The 2019-based line at 54% fertility intersected the points for the 2018 values of 16 Gy male × virgin female egg sterility over time; it also showed similar variance to the 2019 dosimetry shown on the right of the figure. The sterility is probably too variable for an effective SIT program, but there was also no fully sterilizing dose reached (the reported dose of 32 Gy was insufficient here, while Welsh et al. [26] reported that 32 Gy severely reduce mating and subsequent oviposition rates). There was an apparent rise in number of hatching progeny of irradiated males with virgin females for two data points after 40 days (Figure 3), which could be an artefact of the scatter. The higher egg hatch rate in the full irradiated male crosses (54% rather than the expected at 20% hatch from Welsh et al. [26]) was internally consistent with the other experiments with lower hatch than expected here. Notably the 5:1 overflooding treatment with 52% observed hatch where the expected was 62% hatch (Table 3) is in the right direction for success, but could be explained as in the range of control mortality.

The levels of sterility for the irradiation methodology used here produced results which were expected by the internal controls, indicating overall competitive fitness of irradiated males with untreated males. This bodes well for the SIT in BMSB. Differences in sterility levels were observed compared with a previous study, with a higher rate of egg development here at 16 Gy than reported previously (54%, not the 20% egg hatch expected, and it had a much wider spread). This suggests that the variance offered at this facility was too high for SIT, and in contrast with the USDA-ARS supported dosimetry [26]. We predicted a background of wide scatter in 2018, based on the results in the competition experiments and went back to basic dosimetry to find it. Mapping of the radiation field was not available to us, although this was extremely helpful in system design: [37]. The higher baseline hatch in the case of offspring from 16 Gy-treated males (54% hatch in Italy compared with an expected 20% hatch in USA, leading to 99% mortality by F_2_) [26] warrants further investigation, but ultimately requires much greater precision in dosimetry [37]. The same irradiation system reported here has been used on the red palm weevil [38].

Welsh et al. [26] may have different dosimetry outcomes from the results here but this does not account for the high variance, which may have originated from the configuration at ENEA in Rome. The virgin bugs that were used in Experiments 1 and 2 were 5–10 days old at irradiation. Since only the one previous study had been undertaken, with a different irradiator, this 30% difference in expression of sterility from apparently the same dose is unfortunate, but could potentially be explained by a range of factors, including the irradiation right after emergence in USA [26], compared with slightly older males here. It is possible that differences in adult male age at irradiation or stress from transport could have contributed to the difference between studies. Comparative irradiator studies would be beneficial to increase confidence [39,40]. Other irradiators could be available [41]. More dosimetry needs to be examined with a larger number of insects. Using equipment actually intended for treatment before release is, therefore, recommended to sort out the teething problems of this study. In the end, the insects to be released should reflect the same irradiation, handling and physiological conditions as those tested during development. The results using 9 cm vertical cardboard cages for male irradiation showed unexpectedly high variance in sterility, with a mean of 54% hatch at 16 Gy. Dosimetry might be claimed within 3.5% of target in a central beam but the local extinction curve would appear steep to say the least, given the 7–96% sterility range observed after paternal irradiation at 16 Gy. Irradiation systems that are able to provide more robust and reliable low variance sterility are essential for progress, and could be expected to reduce the variance in experiments with overflooding. Despite expectedly low sterility compared with Welsh et al. [26], the absolute values do not detract from our ability to learn from changes in sterility levels seen over time in the treatment with a 5:1 overflooding ratio. Overall, the 5:1 overflooding experiments successfully produced a sterile bias, with six of seven egg batches showing >54% sterility for the first 30 days. Egg sterility levels dropped after 45 days (r^2^ = 0.51), possibly due to mortality of irradiated males, leading to a reduction in the overflooding ratio, favoring mating with remaining surviving untreated males. This is consistent with the adult mortality when fewer males and females were available after 30 days in the cages. A shorter life is expected for irradiated males [26], as it is for other species [30].

Control mortality was initially low but sporadic problems followed in some cages. The expected 80% egg sterility from 16 Gy [26], resulted in a cumulative value from death after the egg stage of 98.5% of adult sterility, and therefore, 80% egg sterility was targeted. Egg sterility could change over time in a mixed case, as surviving males are more likely to be of the wild type, since their longevity is on average greater [26]). This effect could be present in the 5:1 cages, where egg survival jumped after 40 days, with sequential batches. Initial egg sterility levels were enough to suggest that sterile males were competing well for the first month at least. Fresh releases at regular intervals would be logical for any SIT program.

In the case of Hemiptera, there simply are no precedent SIT programs. The challenge of mass rearing such a species has been seen as impossible. We propose wild harvesting and release [42,43] which would appear to be an option as long as the field supply lasts. The strong role for the aggregation pheromone and its potential use in wild harvest also warrants investigation following the recent discovery of effective live traps [44]. However, this enthusiasm is tempered by regulatory concerns that must be addressed prior to moving field-caught, sterilized insects from one jurisdiction and releasing in another. Small-scale wild harvest with collection, irradiation and release for SIT has been done previously, targeting a cockchafer (*Melolontha vulgaris* L., Scarabidae) in valleys in Switzerland. It resulted in local extinction [42]. Another early case with a difficult-to-rear cherry fruit fly, *Rhagoletis cerasi*, involved wild harvest and release of irradiated insects in a small orchard [43].

In response to this change over time in relative performance, it seems logical to initially propose the storage of enough sterile overwintered males for five or so releases at one monthly intervals during the season. This concept, of multiple releases from cold-stored sterile male adults would require refinement, but is based on existing knowledge as the best way to operate SIT in BMSB. The chilled storage of field-collected adults has indicated that suitable methods can likely be developed [45,46]. We expect that sterile cool-stored males will outcompete older wild males but may be equivalent with their own age cohort for a month, requiring a regular interval between releases in a program, potentially for five months of spring-summer.

Wild harvest or mass field-collection of BMSB may be possible, which was demonstrated when the team collected ~11,000 overwintering adults in 2018 (Roselli, unpublished data). This concept is based on the very large aggregations seen in autumn in several countries, including Italy [5]. Bug rearing is difficult and labor intensive, but if wild harvest is feasible, this may not be necessary. For BMSB, it is possible to conceptualize a stepwise process of mass field-collection of pre-diapause adults [39] followed by long-term chilled storage and chilled sorting of diapausing males, with irradiation and a suitable release program.

Assuming that the irradiation dose was very similar (a ^60^Co source was used in both cases), two factors can be considered important to explain the scatter in the results in this work and the differences with results obtained by Welsh et al. [26]: the age of irradiated adults and the setting of the experiments. Differences in the age of irradiated adults: according to Welsh et al. [26], both genders used in their trials were not more than 24 h old, while we irradiated slightly older males. In studies with *Aedes albopictus*, it has been demonstrated that mosquito pupae irradiated at different ages show differences in terms of sterility rates [47]. Consequently, we can consider that the differences in adult age are a possible reason for the decrease of sterility in Experiment 1, compared to the values reported in Welsh et al. [26]. Differences in the method are also present: in Welsh et al. [26], for which single couple mating tests were used. Our tests were set up confining more individuals in the same cage. If the irradiation has an effect on the viability of the sperm of this species, the female could be able to make a cryptic choice of the sperm of different males, prioritizing the “wild type” sperm vs the sterile. Garcia-Gonzales and Simmons [48] demonstrated on the cricket *Teleogryllus oceanicus* that sperm quality plays an important role in determining which male has the advantage when males compete for fertilization; they confirmed that sperm viability is the most important factor in sperm-competition context in insects and perhaps this could be responsible for the scatter and trend in the results in the Experiments 2 and 3 on BMSB.

Aggregation could have an influence on assortativity and the outcome of crosses in the field, but if the spatial release strategy is suitably designed, this may be less of a factor. Optimising strategies for release can take into account a “hot spot” approach if there is effective delimitation [49]. New strategies could be considered, of using the aggregation pheromone, possibly in combination with substrate-borne attractive signals [9], to create deliberate, attractive hotspots on the landscape for wild insects, and then targeting SIT releases at these aggregations. Possible mathematical synergy with egg parasitoids [22,50] means that the potential for the sterile insect technique to be used in conjunction with parasitoids is perhaps very advantageous. The overall feasibility is potentially based on an ability to harvest mass aggregations, which we expect to examine next. More dosimetry above 16 Gy and delivery of male endpoint sterility doses in F_1_ insects with low variance, as well as more in-depth studies on male sperm precedence, are needed, including an evaluation of the effects of irradiation on the viability of sperm. Likewise, further investigation into the relationship between radiation dose and fitness is required to inform and optimise a SIT programme with BMSB [51]. Further, when sterilised bugs are transported for release (e.g., from one jurisdiction to another), it is unclear whether diapause will be reliably broken for ongoing releases, and whether released bugs will maintain fitness post-release, after a rear-out process. We predict greater competitiveness from cool-stored, extended diapause insects, especially against older wild-type males.

## 5. Conclusions

The release of sterile insects offers the potential for population reduction of the pest, with benefits in long term pest management as well as eradication. However, in the case of trans-jurisdictional release, there needs to be a consideration of associated organisms (such phytoplasmas like Paulownia witches broom). The risk of associated organisms would need to be carefully checked [52] to ensure that shipping from one area to another will not introduce pathogens, although the presence of obligate mutualistic gut symbionts may lower this risk [52].

These results suggest that the mating competitiveness and competency of males after irradiation at 16 Gy is not limiting to the sterile insect technique for *H. halys* suppression. Wild harvest of overwintering males using the aggregation pheromone, followed by irradiation and male release, might replace rearing, although regulatory approvals could prove challenging. Mass-collected, sterilized bugs could be transported from an area of high *H. halys* density and released to enable suppression or eradication elsewhere. This concept is under development, but further work is now needed to understand the difference in results between the US and Italian irradiators, and to repeat the dosimetry to consistently demonstrate close-to-full sterility.

## Figures and Tables

**Figure 1 insects-10-00411-f001:**
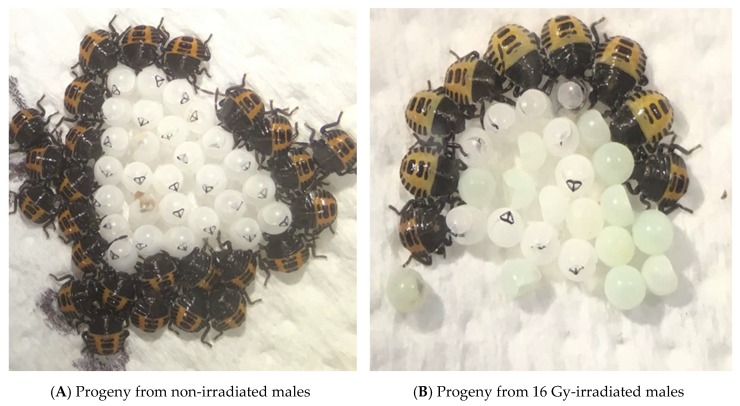
Eggs of *Halyomorpha halys* with a normal complement of hatched nymphs (**A**), and those after male parental irradiation at 16 Gy, with a number of undeveloped, sterile (clear) eggs (**B**).

**Figure 2 insects-10-00411-f002:**
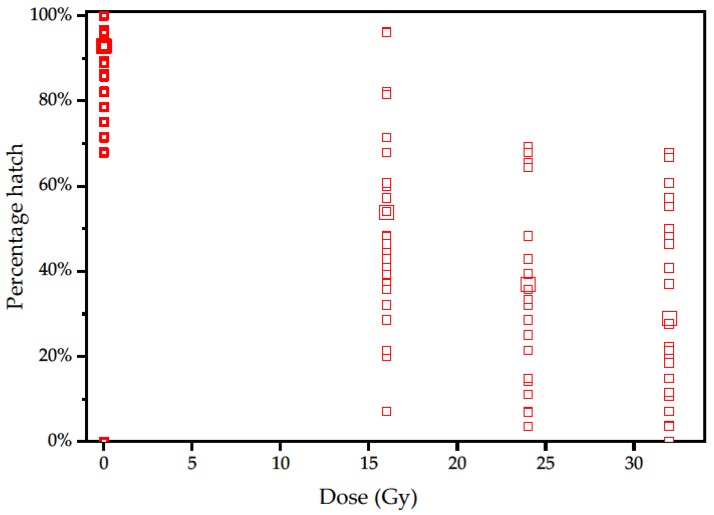
Hatch rate of individual batches of eggs from crosses between irradiated (16, 24 or 32 Gy) or untreated males (0 Gy) mated with virgin females of *Halyomorpha halys*; larger symbols show the mean for each dose.

**Figure 3 insects-10-00411-f003:**
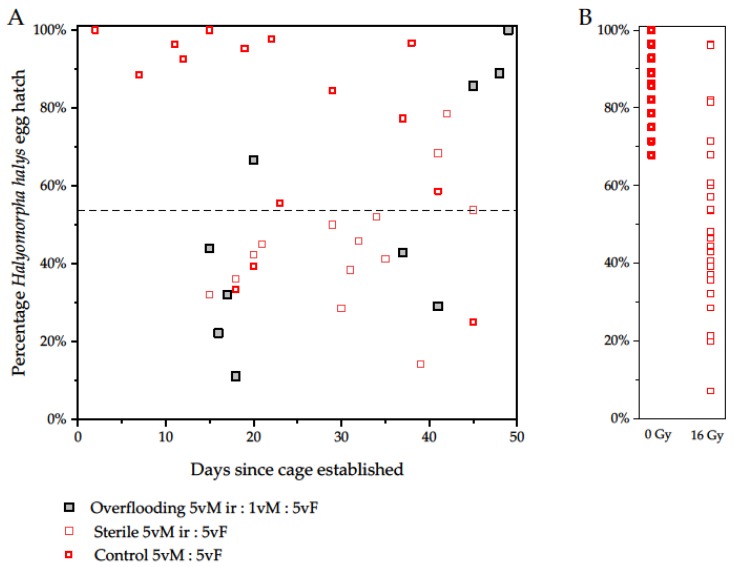
(**A**) *Halyomorpha halys* egg hatch over time as a function of overflooding with a five-to-one ratio of sterile to untreated virgin males (filled grey centre), compared with presumptively fully-sterile × virgin crosses (thin bordered square) and untreated controls (solid bordered square), with 15 virgin females per cage. The line shows the average fertility (54%) for parental male irradiation at 16 Gy. The results on the right (**B**) are from Experiment 2 (Figure 2), showing the spread of egg hatch values from 0 and 16 Gy treatments.

**Table 1 insects-10-00411-t001:** Hatch rate of all batches of eggs from crosses between irradiated (16, 24 or 32 Gy) or untreated males mated with virgin female *Halyomorpha halys*. Eggs per batch have 95% confidence limits. Percentage hatch was corrected after Abbott [29].

Gy	Batches	Eggs Per Batch	Eggs Laid	Hatched	Percentage Hatch	Corrected Hatch
0	58	26 (25–27)	1480	1360	92%	
16	53	25 (23–26)	1319	664	50%	54.4%
24	36	26 (24–28)	853	287	34%	37.0%
32	36	24 (22–26)	871	231	27%	29.4%

**Table 2 insects-10-00411-t002:** The effect of male irradiation on *Halyomorpha halys* egg hatch from five-to-one ratios of irradiated (ir, 16 Gy) and untreated virgin male parents placed with five virgin females, compared with untreated controls and overwintered males with the same treatments.

Treatment	Male Type	Egg Status	Number of Batches	Total Eggs	Observed Hatch	Expected Hatch	*X* ^2^	*p*
Virgin males (5:1)								
16 Gy	5 ir males:1 male:5 virgin females	Laid	21	557	42%	62% ^1^	96.2	<0.0001
		Hatched		233				
Control	5 males:5 virgin females	Laid	4	72	85%	100%		
		Hatched		61				
Overwintered males (5:1)								
16 Gy	5 ir males:1 male:5 virgin females	Laid	26	447	61% ^2^	62% ^1^	0.09	0.75
		Hatched		274				
Control	5 males:5 virgin females	Laid	14	295	94%	100%		
		Hatched		277				

^1^Table 1 provides the estimate of a 54% hatch for 16 Gy irradiated male parents, which rises to a 62% expected hatch with a 5:1 release rate of sterile:wild insects (5/6 × 54% + 1/6 × 100%). ^2^ Significantly different than observed for virgin males at 16 Gy above, *p* < 0.001.

**Table 3 insects-10-00411-t003:** Hatch of *Halyomorpha halys* eggs from oviposition after caging with five-to-one ratios of irradiated (ir, 16 Gy) and untreated virgin male parents placed with five virgin females, compared to untreated controls and undiluted irradiated males (expected values from Table 2).

Treatment Using Virgin Males and Females	Number of Batches	Total Eggs	Eggs Hatched	Observed Hatch %	Expected Hatch %	*X* ^2^	*p*
5 ir males:1 male:5 females	18	398	207	52.0	62 ^1^	16.86	0.0001
5 ir males:5 females	15	373	148	39.7	54 ^2^	30.80	0.0001
5 males:5 females	33	794	646	81.4	100		

^1^Table 2: 5/6 × 54% + 1/6 × 100%, control mortality not included; ^2^
Table 1.

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
