# Peer review of "The Competitive Mating of Irradiated Brown Marmorated Stink Bugs, Halyomorpha halys, for the Sterile Insect Technique"

_insects, 2019, doi:10.3390/insects10110411_

Round 1

Reviewer 1 Report

It was a pleasure to go through your paper titled "Competitive Mating of Irradiated Brown Marmorated Stink Bugs, Halyomorpha halys for the Sterile Insect Technique", even because of the complexity I found the text sometime hard to follow.

I have two main concerns to be addressed during revision:

in several experiments you have mortality occurring in the control. I wonder why not correcting the treatment data with Abbot? which may partially solve some of the problem you faced with comparability of data with previous studies. cardboard containers may not be the best option for irradiation. This is for specific technical reason related with the relation of the container material with gamma rays passing through. This may explain the large variability you have suffered. Please the Gafchromic® Dosimetry System for SIT-IAEA-D4.10.16.

Other suggestions are:

the Results section contain a large part of text that can be conveniently moved to the Discussion.  see comments ncluded into the text

Author Response

Reviewer 1

It was a pleasure to go through your paper titled "Competitive Mating of Irradiated Brown Marmorated Stink Bugs, Halyomorpha halys for the Sterile Insect Technique", even because of the complexity I found the text sometime hard to follow.

The revision has added clarity we believe.

I have two main concerns to be addressed during revision:
in several experiments you have mortality occurring in the control. I wonder why not correcting the treatment data with Abbot? which may partially solve some of the problem you faced with comparability of data with previous studies.

The data have now been corrected using Abbott (1925), and this has lead to a revision of the Chi-squared statistics, but in fact there was only minor change to significance levels.

Cardboard containers may not be the best option for irradiation. This is for specific technical reason related with the relation of the container material with gamma rays passing through. This may explain the large variability you have suffered. Please the Gafchromic® Dosimetry System for SIT-IAEA-D4.10.16.

We have reconsulted the Italian Atomic Energy Agency who conduct their calculations and now cite their bulletin which is available about this system. We have revised the text in this area. We consulted the SOP recommended above.

Other suggestions are:
the Results section contain a large part of text that can be conveniently moved to the Discussion.

This has been done

  see comments included into the text

This text has been moved and the text extensively revised.

Reviewer 2 Report

This manuscript titled ‘Competitive mating of irradiated brown marmorated stink bugs, Halyomorpha halys for the sterile insect technique’ provides information on the sterility of irradiated males at 0,16, 24 and 32 Gy.

The results show that there was 50 % sterility of eggs at 16 Gy. This study provides preliminary or fundamental data to evaluate the potential application of sterile insect technique in suppressing the BMSB populations.

The article is well written and easy to understand. Methods are explained in detail. The authors can also mention what was the average number of eggs in each egg mass. For example, in Figure 1 the number of eggs differ in the two treatments. Mention the full form of Gy (Gray?), when you first talk about it in the manuscript.

Specific comments are listed below:

Line 72 – Delete ‘)’ after [25]

Line 147 – The size of the figure and the labels can be reduced

Line 242 – The y-axis of figure 3 is cut off. The size of the percentage egg hatch scale and the labels can be reduced

Author Response

Dear Editors

Thank you for the opportunity to revise this manuscript. We have accepted the reviewers comments in full in order to improve the manuscript. The details follow.

Yours sincerely,

DM Suckling

Reviewer 2

Comments and Suggestions for Authors
This manuscript titled ‘Competitive mating of irradiated brown marmorated stink bugs, Halyomorpha halys for the sterile insect technique’ provides information on the sterility of irradiated males at 0,16, 24 and 32 Gy.
The results show that there was 50 % sterility of eggs at 16 Gy. This study provides preliminary or fundamental data to evaluate the potential application of sterile insect technique in suppressing the BMSB populations.
The article is well written and easy to understand. Methods are explained in detail.

The authors can also mention what was the average number of eggs in each egg mass. For example, in Figure 1 the number of eggs differ in the two treatments.

   These data are now added to Table 1.

Mention the full form of Gy (Gray?), when you first talk about it in the manuscript.

   Done

Specific comments are listed below:
Line 72 – Delete ‘)’ after [25]

   Done

Line 147 – The size of the figure and the labels can be reduced

   Figures redone.

Line 242 – The y-axis of figure 3 is cut off. The size of the percentage egg hatch scale and the labels can be reduced

   Figures redone

Round 2

Reviewer 1 Report

I am now satisfied with the revisions you provided. Thanks